# Biology of the Extracellular Proteasome

**DOI:** 10.3390/biom12050619

**Published:** 2022-04-21

**Authors:** Gili Ben-Nissan, Naama Katzir, Maria Gabriella Füzesi-Levi, Michal Sharon

**Affiliations:** Department of Biomolecular Sciences, Weizmann Institute of Science, Rehovot 7610001, Israel; gili.ben-nissan@weizmann.ac.il (G.B.-N.); naama.katzir@weizmann.ac.il (N.K.); maria.fuzesi@weizmann.ac.il (M.G.F.-L.)

**Keywords:** body fluids, proteasome, protein degradation, extracellular vesicle, pathophysiology

## Abstract

Proteasomes are traditionally considered intracellular complexes that play a critical role in maintaining proteostasis by degrading short-lived regulatory proteins and removing damaged proteins. Remarkably, in addition to these well-studied intracellular roles, accumulating data indicate that proteasomes are also present in extracellular body fluids. Not much is known about the origin, biological role, mode(s) of regulation or mechanisms of extracellular transport of these complexes. Nevertheless, emerging evidence indicates that the presence of proteasomes in the extracellular milieu is not a random phenomenon, but rather a regulated, coordinated physiological process. In this review, we provide an overview of the current understanding of extracellular proteasomes. To this end, we examine 143 proteomic datasets, leading us to the realization that 20S proteasome subunits are present in at least 25 different body fluids. Our analysis also indicates that while 19S subunits exist in some of those fluids, the dominant proteasome activator in these compartments is the PA28α/β complex. We also elaborate on the positive correlations that have been identified in plasma and extracellular vesicles, between 20S proteasome and activity levels to disease severity and treatment efficacy, suggesting the involvement of this understudied complex in pathophysiology. In addition, we address the considerations and practical experimental methods that should be taken when investigating extracellular proteasomes. Overall, we hope this review will stimulate new opportunities for investigation and thoughtful discussions on this exciting topic that will contribute to the maturation of the field.

## 1. Introduction

Decades of research have focused on the intracellular proteasome complex, yielding detailed information on the function, structure and regulation of this proteolytic machinery (see reviews [1,2]). Nevertheless, it is becoming clear that assembled and functional proteasome complexes also exist in extracellular body fluids. The first discovery of extracellular proteasomes occurred about 30 years ago when the presence of the proteasome in the blood was reported [3]. At the same time, the proteasome was also identified as one of the extracellular ascidian (sea squirt) sperm proteases involved in fertilization [4], a finding which was later on corroborated in mammals [5]. Since then, clinical studies have indicated the presence of the complex in various extracellular locations, among them extracellular vesicles (EVs) [6], cerebrospinal fluid [7], bronchioalveolar lavage [8], epididymal fluid [9] and plasma [10]. This diversity of extracellular environments suggests that proteasome-dependent protein quality control also operates in the extracellular milieu of multicellular organisms.

The extracellular fluid comprises approximately 20% of total body weight, and is mainly subcategorized into interstitial fluid that fills the spaces between cells (~12% of body weight), plasma (~5% of body weight) and transcellular fluid that is contained within epithelial lined spaces (~1.5% of body weight, e.g., cerebrospinal fluid, bladder, aqueous humor in the eye and serous fluid) [11]. The composition of the body fluids is dependent on which region or organ of the body contains the fluid [12]. Plasma and interstitial fluids are similar in composition, containing high concentrations of sodium, chloride, bicarbonate and proteins, but relatively lower in potassium, magnesium and phosphate. Physiologically, interstitial fluids tend to display lower concentrations of proteins. Moreover, various clinical pathologies can alter the fluid composition and its constituents in the multiple compartments of the human body [13]. This phenomenon is highly relevant to this review, as extracellular levels of the proteasome complex have been shown to increase in certain disease states [10]. 

The proteasome comprises a conserved degradation machinery that is vital for maintaining proteostasis, as it determines the abundance of each protein within the proteome, prevents the accumulation of damaged or misfolded proteins and controls the levels of short-lived regulatory proteins [14]. Two alternative proteasomal degradation mechanisms, which are not mutually exclusive, exist in cells. The first degradation route is dependent on ubiquitin and ATP, while the second functions independently of either of these factors [15,16]. The 26S proteasome, which mediates the ubiquitin and ATP-dependent degradation route, comprises a 19S regulatory particle, which recognizes ubiquitin-tagged substrates, and a 20S catalytic core particle, where substrates are degraded via the breakage of peptide bonds (Figure 1). Degradation by the 26S proteasome is coordinated by three different types of ATP-dependent enzymes (E1, E2 and E3) that ubiquitinate the substrate and sensitize it to degradation [17]. In contrast, the 20S proteasome, acting alone, can degrade protein substrates in a ubiquitin- and ATP-independent manner, by cleaving unfolded or unstructured regions within its substrates [15,18]. 

Ubiquitin- and ATP-independent degradation can also occur through the binding of the 20S proteasome to alternative activators such as PA28αβ, PA28γ and PA200 (Figure 1) [19,20]. These regulators open the 20S proteasome gate and enhance the complex’s catalytic activity; however, unlike the 19S complex, they function in an ATP- and ubiquitin-independent manner. The recruitment of these proteasome activators to the 20S particle enables the rapid adaptation of proteasome function in response to cellular stimuli, as they may influence substrate selection and affect product outcome due to allosteric effects on β-catalytic active sites [21]. Nevertheless, the free 20S proteasome, when liberated from its activators, represents the most abundant fraction of the total proteasome content in many cell types [22,23].

The 20S proteasome degrades proteins through three proteolytic processes performed by the β-subunits PSMB5, PSMB6 and PSMB7, which display chymotrypsin (ChL)-, caspase (CasL)- and trypsin (TrpL)-like activities, respectively [15,18,24,25]. Immunoproteasomes are generated by replacing these three catalytic subunits with their counterpart subunits PSMB8, PSMB9 and PSMB10, respectively [26]. This leads to the generation of different cleaved peptides, in comparison to those of the standard proteasome [27]. Immunoproteasome subunits are constitutively expressed at low levels in various cell types, and to a greater extent in immune and lymphatic tissues in response to inflammatory cytokines, e.g., interferon-γ and TNF-α [28]. In addition, they are upregulated and incorporated into newly assembled 20S proteasomes in various tissues in response to specific stimuli such as cytokine treatment, viral [29], fungal or bacterial infection, injury, ischemia [30], muscle atrophy due to aging [31] or diabetes [26]. Intermediate proteasomes containing a mixture of both the standard and immune subunits have also been described [32,33]. In addition, diversification of individual subunits due to genetic variations, alternative splicing and numerous post-translational modifications lead to the generation of multiple distinct proteasome species [22]. This heterogeneity likely encodes key functional features, resulting in rapid response and adaptation to varying biological conditions.

Knowledge of extracellular proteasomes is lagging well behind that of their intracellular counterparts. Not much is known about the origins and mechanisms of secretion of the complex, putative substrates and regulatory modes. Moreover, it is still not clear whether the free 20S proteasome is present in the extracellular space, whether this core particle is bound to the 19S particle forming the 26S proteasome or if it associates with its alternative activators, PA28αβ, PA28γ and PA200. It is also possible that different configurations of the complex exist in the various extracellular locations. In addition, it is possible that outside of cells, proteasomes act differently than their intracellular counterparts. Given the early stages of research in this field, our purpose here is to bring forward the information that has accumulated to date on extracellular proteasomes, and to highlight current gaps in knowledge. In doing so, we will provide an integrated overview of the complex’s presence in 25 different extracellular body fluids, and discuss in detail two of these compartments, blood and EVs, the only sites that were more thoroughly studied. Furthermore, we address factors that should be taken into consideration when investigating this complex, and possible experimental methods and tools that may facilitate developments in the field. Overall, we hope this review will encourage researchers to design extracellular proteasome-related projects, and contribute to the maturation of this exciting field. 

## 2. Integrated Analysis of Proteomic Data from Body Fluids

To gain a comprehensive understanding of the potential presence of the proteasome in body fluids, we took advantage of the large datasets generated by proteomic analysis. Overall, we screened 143 different proteomic datasets that have been accumulating over the last 16 years (Appendix A). Our screen revealed that proteasome components are present in at least 25 different extracellular body fluids (Figure 2), and in 88% of the overall datasets (Figure 3A). In addition to the major body fluids, namely blood and urine, these also include secreted tears and sweat, as well as other more microscale and difficult to extract fluids such as saliva, cerebrospinal fluid, pericardial fluid, seminal fluid, amniotic fluid, cervicovaginal fluids, milk, nipple aspired fluid, exhaled breath condensate, bile, lymph fluid, cerumen and pancreatic juice. Moreover, proteasome constitutes are found in various disease-related fluids such as ascites, synovial fluid, sputum, bronchoalveolar lavage, lymphatic drainage, pancreatic cyst fluids, lung cancer pleural effusions and various cancer interstitial fluids. The extracellular presence of the proteasome is also recapitulated in cultured cells, where particles are found in the growth medium [6,34,35,36,37]. Remarkably, in over 48% of these extracellular fluids (Figure 2), comprising 24% of the datasets, proteasomes were also detected in extracellular vesicles found within (Figure 3B). 

The presence of individual proteasome subunits does not necessarily indicate the existence of an active and assembled 20S complex. It can also reflect the existence of unassembled subunits or residual contamination from surrounding cells that were not cleared during sample collection and/or preparation. However, none of the examined proteomic studies of body fluids investigated this aspect (Appendix A); namely, verification of the integrity and functionality of the complex were not undertaken (see suggested methods below). Therefore, we relied on a threshold value of at least 6 subunits out of the 14 to define the presence of the 20S proteasome. Our rationale was that the presence of at least 40% of the complex’s building blocks likely indicates the existence of the intact particle with reasonable confidence. Overall, within the 143 datasets examined, 104 passed the threshold bar, representing an impressive variety of body fluids (Figure 2). This observation emphasizes that the extracellular presence of the proteasome is not a rare event, but rather reflects a general phenomenon seen in body fluids.

Within the 104 datasets containing the 20S proteasome, 72 studies (69%) reported the presence of at least one immunoproteasome subunit (Figure 3C). In general, the relative amounts of immunoproteasome in the human body vary substantially between organs and can range from almost none, e.g., in the brain and muscles, to 70% of the total 20S proteasome population in the lungs, bone marrow, thymus and spleen [38]. In extracellular fluids, immunoproteasome enrichment was reported in bronchoalveolar lavage (4/4 datasets) and in lung cancer pleural effusions (7/9 datasets), which originate in the lungs. Immunoproteasomes were also enriched in breast milk (4/5 datasets), in the synovial fluid (3/3 datasets), tears (5/6 datasets), saliva (9/10 datasets), cerebrospinal fluid (4/7 datasets) and blood (5/7 datasets). Overall, the widespread presence of the immunoproteasomes suggests that together with the standard proteasome, they facilitate physiological processes.

Of the three immunoproteasome subunits, PSMB8 predominates, appearing in 83% of the studies identifying immunoproteasomes, either on its own (17%), together with PSMB9 (29%) or with both PSMB9 and PSMB10 (33%). The remaining datasets identified less common intermediate proteasomes such as PSMB9 or PSMB10 alone, PSMB8 together with PSMB10 or PSMB9 together with PSMB10. These results are in line with studies on cellular immunoproteasomes where PSMB8 is also the predominant proteasome immunosubunit [22,38], suggesting that the overall cellular 20S proteasome composition is mirrored in the extracellular milieu. The thymoproteasome β subunit (PSMB11) was not detected in the datasets examined, and the spermatoproteasome α subunit (PSMA8), as expected, was identified in two studies focusing on seminal fluids. 

A major unknown in the field concerns the composition of the proteasome complex in body fluids; namely, whether the 20S proteasome functions in its free form, or whether it associates with its activator(s), and if so with which one(s). Here, we found that in 27% of the datasets examined, only 20S proteasome subunits were identified, suggesting that it acts in its free form. In 70% of the studies, the 20S proteasome was accompanied by PSME1/2 subunits that form the PA28α/β regulatory complex, either with or without additional 19S subunits (Figure 3D). The PSME3 (PA28γ) and PSME4 (PA200) proteasome regulators, on the other hand, appeared in only 9% and 7% of the datasets, respectively, (of those, in 3% they appeared together), and only when accompanied by the PA28α/β complex. Notably, the proteasome inhibitor PSMF1 (PI31) was quite abundant, being present in 19% of the datasets, and almost exclusively together with PA28α/β (except for one dataset). In regard to the 26S complex, we set a distinct threshold value for the two subcomplexes that form the 19S particle, the lid and base. To define the presence of the base (encompassing 6 subunits), the bar was set to a minimum of three ATPase subunits, while for the lid (composed of the 12 core subunits, PSMD1–8 and PSMD11–14) the threshold was set to a minimum of six subunits. Based on this threshold, the 19S base and lid subunits were identified together with the 20S proteasome in 23% of the datasets. Interestingly, with the exception of one dataset, the 19S base and lid components always appeared together with PA28α/β. In 6% of the studies, only 19S base components were identified together with the 20S proteasome and PA28α/β, and in 7% of the studies, only the 19S lid components were identified with the 20S proteasome, although still together with PA28α/β; however, the existence of lid components without the 19S base is probably insignificant. Overall, it seems that the PA28α/β activator was found in the vast majority of the body fluids examined (except for lymph fluid and sweat), indicating that it plays a fundamental role in activating extracellular 20S proteasomes, and possibly in hybrid form, together with the 19S proteasome regulator. 

The number of studies that have specifically examined proteasome integrity and configurations in extracellular body fluids and confirmed the exclusive existence of one type of proteasome and not the other is very limited. Studies that did put an emphasis on this aspect mostly suggest that only the 20S proteasome, rather than 26S particles, is present in the extracellular milieu [8,10,39]. Nevertheless, in some studies, the presence of 26S proteasomes was also noted, such as in the alveolar space of the injured lungs [40] and in bronchoalveolar lavage fluid [41]. Remarkably, as far as we know, the biochemical verification of the intact 20S/PA28α/β complex in the extracellular milieu has not been demonstrated. Overall, it is likely that generalizations cannot be made regarding the extracellular proteasome composition; rather, each body fluid needs to be investigated separately in order to understand the specific involvement of the extracellular proteasome system.

In addition to freely flowing proteasomes, the complex was also found to be encapsulated within extracellular vesicles (EVs). Specifically, in our analyses (29 studies altogether), proteasomes delivered within EVs were identified in 12 of the 25 body fluids. While it can be argued that some proteasome components may be present in total body fluids due to leakage from the surrounding cells during sample collection or preparation, it is less likely that proteasomes enclosed within EVs result from contamination. However, it should be noted that while EVs were specifically isolated from the body fluid in some of the proteomic studies, other studies have left a shadow of ambiguity regarding their presence.

In body fluids, EVs are typically present in relatively low amounts. Consequently, their overall protein content is low, whereas purification of a sufficient amount of EVs for high quality proteomic analysis requires large volumes of body fluids, which are not always available. Nevertheless, in all 29 EV-based studies, we identified 20S proteasome subunits, and in 25 of these studies, six or more 20S proteasome subunits were identified, indicating that the proteasome indeed constitutes an integral part of the EV proteome in body fluids. We found that within these 25 EV-based studies, 20 datasets (80%) contained immunoproteasome subunits, a value that is higher than the abundance of immunoproteasomes in the overall body fluids (69%, Figure 3C), suggesting that these unique proteasome species are specifically delivered by EVs. As found for the extracellular compartments, the majority of the EV studies identified PA28α/β (76%) (Figure 3D); 24% of the datasets contained subunits of just the 20S proteasome; 28% of the datasets identified the 20S proteasome together with the 19S base, lid and PA28α/β subunits; while all these proteasome components together were identified in only 22% of the entire body fluid datasets (Figure 3D). Thus, not only the type of body fluid indicates the proteasome configuration, but also its packing and whether it circulates freely or is encapsulated within vesicles.

The E1, E2 and E3 ubiquitination enzymes as well as ubiquitin were present in 18 of the 29 EV studies and in 72% of the body fluid datasets. However, the feasibility of robust ubiquitination cycles in body fluids seems less likely, given the low ATP concentrations in these compartments. Accordingly, several studies on urine and blood, which quantified the protein’s relative abundance, indicated comparable levels of 20S subunits and PA28α/β, while the levels of 19S proteasome subunits were about 10-fold lower [19,42,43,44,45]. In two urine EV studies, on the other hand, the levels of 20S, 19S and PA28α/β subunits were found to be relatively similar [45,46]. A dedicated analysis of each body fluid that entails separation between free and encapsulated proteasomes and determination of the relative abundance of each proteasome configuration will advance our understanding of the extracellular proteasome system. 

## 3. Plasma Circulating Proteasomes

Proteasomes circulating freely in the plasma were the first type identified in the extracellular space [3]. This probably explains the large amount of information that has been gathered so far on these complexes in comparison to proteasomes present in other body fluids. To date, this is one of the few extracellular compartments in which assembled and active complexes were denoted, in contrast to other body fluids in which only the presence of proteasome subunits was identified (Figure 2). Nevertheless, much knowledge is lacking concerning the nature of plasma proteasomes, as their origins, mode of regulation, composition and substrate specificities in physiology and pathology remain unclear.

The integrity and activity of the circulating proteasomes was first demonstrated about 15 years ago by a study that isolated proteasomes from plasma of healthy donors and from patients with autoimmune disease [47]. The 20S proteasome particles were found to display the same size and shape as erythrocyte proteasomes, and to harbor both standard and immune subunits. Further validation of the presence of free uncapped 20S proteasomes in plasma, rather than 26S complexes, came from more recent studies using enzyme-linked immunosorbent assays (ELISA) [48] and native PAGE analysis [49]. On the other hand, proteomic analyses of plasma samples identified standard and immune subunits of the 20S particle, as well as subunits of the 19S, PA28αβ and PA200 proteasome activators (Figure 1) [50,51,52,53,54,55]. However, it should be noted that these studies did not examine whether the proteasome activator subunits were not just present but actually assembled and active. Moreover, the vast majority of the experimental workflows in these studies did not distinguish between proteasomes freely flowing in the blood and those encapsulated within EVs. We suggest that future studies should apply more rigorous criteria in order to understand whether or not proteasome regulators are really bound to plasma proteasomes, and if there are differences in the nature of free and encapsulated plasma complexes.

So far, the origins of the circulating proteasomes remain enigmatic. Proteasome subunits do not contain transport signals, and no specific transporter is known to facilitate the exit of such large complexes through the plasma membrane. Moreover, chromatographic analysis of 20S proteasomes isolated from erythrocytes, thrombocytes, monocytes and T cells yielded a different elution profile than that of circulating complexes, suggesting that blood cells are not the major source of circulating proteasomes [47]. Cytolysis or defective apoptosis of blood or endothelial cells was also dismissed as a possible source for circulating proteasomes, as a poor correlation was found between the indicator of cytolysis and cell death, i.e., lactate dehydrogenase (LDH), and levels of the 20S proteasome [56,57,58,59]. Another possibility that was raised is that proteasomes are released into the plasma by the breakdown of EVs. This notion is supported by the fact that active 20S proteasomes are present within EVs [39,60] and that shedding of EVs into the plasma has been documented [61,62]. Thus, EVs may serve as a short-lived shuttling and storage compartments for 20S particles prior to their liberation into the plasma. If this is indeed the case, pharmacological modulation of the EV secretion system may influence circulating 20S proteasome levels, thereby possibly affecting disease etiology and therapeutic capabilities.

Multiple clinical investigations have focused on the potential use of the circulating proteasome as diagnostic markers. This is due to the correlation that has been found between elevated levels of 20S proteasomes in human serum and multiple pathological conditions, including aggressive hematologic malignancies, autoimmune diseases, inflammatory liver disorders, carcinomas, melanoma, sepsis, surgeries, burns, inhalation injuries and anti-neutrophil cytoplasmic antibody (ANCA)-associated vasculitis (see [10,63]). In lymphoid malignancies such as chronic lymphatic leukemia, multiple myeloma and non-Hodgkin’s lymphoma, decreased levels of plasma 20S particles are detected in the early stages of the disease in comparison to healthy individuals, which vastly increase throughout later, more aggressive disease stages [57,64,65,66]. Besides defining the complex concentration, all three types of 20S proteasome enzymatic activities were found to be positively correlated with the progression of hemathological malignancies [64,65]. ChL activity was found to increase in the plasma of patients undergoing open surgery and laparoscopic surgery [67], in correlation with increased oxidative stress during these surgical interventions. Experiments in mouse models suggested that elevated 20S proteasome levels and activity can serve as predictors of early-stage Alzheimer disease progression [49]. In contrast, reduced circulating proteasome activity in patients with chronic tinnitus and mild cognitive impairment may be specifically linked to cognitive decline in these patients [49]. Taken together, changes in plasma 20S proteasomes activity or levels, when compared to healthy controls, may be adapted in the future to monitor the disease trajectory and to design biomarker-guided targeted therapies. 

## 4. Proteasomes Encapsulated within Extracellular Vesicles 

With the rising interest in extracellular vesicle content, worldwide proteomic initiatives have begun to expose the composition of their protein cargo. An interesting observation arising from the accumulated data is that proteasome subunits are secreted within EVs found in both body fluids [45,55,68,69,70,71,72,73,74,75] and in the growth medium of mammalian cells [6,34,36,37,76]. These proteasome-containing vesicles are heterogeneous in terms of size (ranging between 30 and 500 nm in diameter), cellular compartment source, cellular origin, target destination and content, suggesting that their functions are also varied [34,77,78,79]. In addition to the 20S proteasome subunits, various studies also identified the 19S and PA28αβ subunits in the secreted EVs (Figure 3D). Apparently, the PA28αβ regulator was found to be the major proteasome activator within EVs, suggesting that as in the fluids themselves, interactions of the EV proteasomes with this regulator plays a significant role, possibly enhancing ubiquitin- and ATP-independent degradation, thereby reducing its reliance on an enzymatic ubiquitination cascade and energy costs.

Proteomic knowledge constitutes the main source of data regarding the presence of proteasomes within the various secreted vesicles. The identification of proteasome subunits does not necessarily indicate that an assembled and active complex is delivered. However, it is challenging to perform extensive biochemical studies on EV proteasomes in human body fluids due to sample availability, reproducibility and ethical issues. Consequently, only a limited number of studies have specifically investigated this aspect of proteasome activity. While studies on functional proteasomes within body fluid EVs are scarce, research has mostly been conducted on EVs secreted by cells in culture [37,80,81], which pose a reasonable alternative source for biologically relevant EVs. Nevertheless, active proteasomes have been identified, for example in circulating exosome-like vesicles purified from mouse serum. Interestingly, proteasomal activity within these vesicles increased following acute kidney injury and femoral arteriectomy, resulting in hind limb ischemia [6]. In another mouse model, all three proteasomal catalytic activities were detected in EVs secreted from blood platelets and inhibited by the proteasome inhibitor epoxomicin [82]. These platelet EVs were shown to process ovalbumin and present one of its unique peptides through MHC-I antigen presentation molecules. Moreover, platelet EVs were shown to support the proliferation of T cells in response to exposure to ovalbumin, a process which was inhibited when the EVs were pre-incubated with epoxomicin. These findings demonstrate that platelet EVs contain functional proteasomes, as well as the required machinery to process and present antigens to CD8+ T cells. In these cases, however, it is not entirely clear whether the proteasome functions as an intact 20S catalytic core or is bound to one of its activators.

In cell cultures, the levels and proteolytic capacities of EV-derived proteasomes were shown to depend on the cells from which they were secreted [37]. Even within the same cell culture, proteasome levels and composition may vary between small and large vesicles (often referred as exosomes and microvesicles, respectively), as identified by biochemical Western blot analysis [83] and quantitative proteomics [34]. Moreover, in body fluids, the levels of proteasomes enclosed within EVs were altered in different diseases in comparison to EVs from healthy individuals, suggesting that EV proteasomes could serve as useful biomarkers for assessing disease severity and progression. 

For instance, a recent proteomic study comparing exosome cargoes from serum samples of gastric cancer patients with those of healthy individuals revealed that the majority of 20S proteasome subunits were significantly enriched in vesicles collected from patients in comparison to those from healthy individuals, a result which was also confirmed by Western blots in a separate validation cohort [84]. Interestingly, the levels of 20S proteasome subunits were found to be particularly upregulated in metastatic gastric cancer exosomes, while no significant difference was found between healthy controls and non-metastatic gastric cancer exosomes, suggesting that EV proteasomes can serve as biomarkers for metastatic gastric cancer and even pose as therapeutic targets in this disease [84]. Similarly, the levels of EV-20S proteasome subunits were increased in plasma samples of colorectal cancer patients, compared to colorectal polyps patients [85], as well as in plasma EVs from breast cancer and ascites EVs from ovarian cancer patients, in comparison to healthy individuals [86].

Proteasome subunits were also found to be differentially regulated in neurodegenerative diseases. For example, the levels of the 20S proteasome subunit PSMA7 were elevated in urine EVs from Parkinson’s disease patients in comparison to neurologically normal controls [87]. On the other hand, 20S proteasome subunits were found to be downregulated in cerebrospinal-fluid-derived EVs from amyotrophic lateral sclerosis (ALS) patients [88]. Taken together, these results imply that proteasome levels in EVs depend on both the distinct body fluid being examined and the extent of disease.

To date, it is not clear how proteasomes are targeted to EVs, nor their specific function within the EV context. EVs are thought to serve as an alternative shuttling mechanism to exocytosis [89]. As such, they facilitate cargo delivery to recipient cells by several mechanisms, including endocytosis, fusion to the plasma membrane or phagocytosis [90]. Thus, it is reasonable to assume that EV proteasomes execute their function following introduction into target cells. However, EVs may also facilitate the export of proteasomes outside of cells, releasing the complex into surrounding body fluids to act on extracellular target substrates. However, for this to happen, the content of the EVs should be emptied into the extracellular milieu. In this context, it was shown that lipid-degrading enzymes such as secretory phospholipase A2 and sphingomyelinase, were able to dismantle EVs and consequently liberate proteasomes into the extracellular environment [62]. Secretory phospholipase A2 typically functions in the extracellular space, and its levels are found to increase in diseases such as cancer, atherosclerosis, immune disorders and inflammation [91]. In line with these results, pro-inflammatory mediators were shown to facilitate the release of lipid-degrading enzymes in response to cytokine activation, e.g., IL-1, INFγ and TNF-α [92], suggesting that these factors may also act to increase the pool of freely floating proteasomes in body fluids in instances of disease. 

Overall, it is thought that EV proteasomes function either in the vicinity of the cells of origin or through circulation within the bloodstream to other parts of the body. Support for the circulation of EVs comes from studies on the therapeutic role of mesenchymal stem cell exosomes, which contain functional and active 20S proteasomes. It was found in a mouse model for myocardial ischemia and reperfusion injury, that intravenous injection of these exosomes reduced the degree of damaged heart tissue [93]. Interestingly, treatment of mice following myocardial ischemia or reperfusion injury with exosomes resulted in accumulation of significantly lower amounts of misfolded proteins or oligomers in the injured heart tissue when compared to treatment with saline [80]. Misfolded proteins are major substrates of the 20S proteasome, which primarily degrade proteins containing unstructured regions. This result, therefore, implies that 20S proteasomes within exosomes may have a therapeutic role during myocardial ischemia or reperfusion injury, possibly by decreasing the load of damaged proteins and reducing chances of protein aggregation in the injured heart tissue. Another study demonstrated that malaria-infected human red blood cells secrete EVs containing assembled and functional 20S proteasome complexes [39]. The EV-20S proteasomes delivered within the bloodstream were shown to modulate the mechanical properties of naïve RBCs by degrading cytoskeletal proteins. This mechanism reduces the stiffness of the membrane and primes naïve red blood cells for parasite invasion, eventually increasing the parasite’s growth [39]. Given the broad range of parasites whose EVs contain proteasomes [94], this process of hijacking the cellular EV system for delivering proteasomes to facilitate the parasites invasion, growth and development, may be more widespread.

EVs encapsulate a range of biologically active cargo, including different RNA molecules, lipids and proteins. As such, they are involved in multiple processes important for disease prevention, but on the other hand also promote cancerous processes through various means, such as altering signaling pathways, enabling malignant cells to escape apoptosis, promoting metastasis, aiding in angiogenesis, evading immune responses, inducing therapeutic resistance, altering the tumor microenvironment and increasing invasive and metastatic activity (reviewed in [95,96,97,98]).

To date, evidence for the roles of EV-encapsulated proteasomes in cancer is mostly indirect. For example, naïve EVs collected from primary multiple myeloma cells and from a human multiple myeloma cell line were shown to stimulate multiple myeloma cell growth [99], as well as endothelial cell migration and proliferation; however, pre-treatment of the primary multiple myeloma cells with the proteasome inhibitor bortezomib repressed these processes. Moreover, the EV proteasomal activity of bortezomib-treated cells was about 50% lower than that seen in naïve EVs, suggesting that the catalytic activity of the proteasome in EVs is involved in regulating angiogenesis and in the proliferation and migration of endothelial cells [100]. In these studies, however, it was not established whether EVs from the bortezomib-treated cells contained a lower number of proteasomes, or whether their catalytic activity was reduced by the inhibition of the proteasomes in their cells of origin. Another study compared the protein content of EVs isolated from serum samples of young female breast cancer patients with those of healthy controls, and demonstrated that proteasome subunits were enriched in the cancerous EVs. Interestingly, cancerous EVs significantly increased cellular invasion in scratch wound invasion assays compared to EVs from healthy donors [101]. Although the specific role of the proteasome was not examined in this process, the fact that proteasomes are found primarily in cancerous EVs points to their possible role in breast cancer metastasis.

How EV proteasomes influence these processes remains unclear. In the event that EV proteasomes are delivered and inserted into target cells, they may execute various proteolytic functions affecting the protein balance and homeostasis at their destinations. For example, EV proteasome-mediated degradation may change the levels of regulatory proteins, consequently affecting signal transduction cascades. On the other hand, if released into the extracellular matrix, proteasomes may function in entirely different ways. One possibility is that EV proteasomes, once released from the vesicle, participate in the degradation of extracellular matrix proteins as part of the extracellular protease arsenal. As has been shown for other extracellular proteases, this activity results in a tumor microenvironment that is less rigid and more prone to invasion [102]. Alternatively, extracellular proteasomes may be involved in the shedding of cancerous cells from the main tumor into the bloodstream, promoting the development of metastasis in an as-yet unidentified manner. Moreover, as all malignant tissues suffer from some degree of oxidative stress [103], it might be that EV 20S proteasomes are delivered to cancer-associated cells to target the growing population of damaged proteins, either inside or outside of the cells.

## 5. Experimental Strategies for Investigating Extracellular Proteasomes

An insight that clearly stands out from Figure 2 is that the integrity and activity of extracellular proteasome complexes has been hardly ever studied. Obviously, this is due to the fact that most of the data originates from proteomic studies that did not specifically focus on the proteasome complex, but rather on the systematic analysis of body fluid proteomes. However, now that a comprehensive map of the extracellular localization of proteasome has been generated, the time is right to deepen our understanding of this extracellular entity using biochemical tools. In practice, features such as the following should be examined (see Figure 2, panel labels): (i) characterizing the integrity of the complex, verifying the presence of an intact proteasome complex rather than disassembled subunits; (ii) determining the configuration of the proteasome complex, i.e., whether a free 20S complex is present, or bound to an activator(s); (iii) analyzing the functionality of the three enzymatic activators ChL, CasL and TrpL; (iv) inhibition of proteolytic activity by specific proteasome inhibitors; (v) differentiating between cell-free proteasomes and those enclosed in EVs. To facilitate these types of analyses, we offer below some experimental factors and methodologies that may be considered when examining extracellular proteasomes.

An initial step that is crucial for the handling of body fluids, but not always performed following sample collection, is centrifugation to remove cells, insoluble materials and debris from the fluid. Application of the appropriate force during sample clearance is particularly significant when analyzing body fluids that are rich in cellular content, e.g., blood and nasal secretions [104]. Insufficient centrifugation may not efficiently remove these cells, resulting in contamination of the sample with cellular proteins, while high centrifugation forces can result in leakage of the cellular content from these cells into the extracellular fluid, depletion of EVs carrying proteasomes and sedimentation of soluble proteins such as proteasomes from the biological fluid [105]. Therefore, when handling biological fluids, a series of differential centrifugation steps is required (Figure 4). 

A recommended setup for differential centrifugation should include an initial step of clearing the sample from live and dead floating cells at 300× *g* (10 min at 4 °C), followed by centrifugation of the supernatant at 1000–2000× *g* (20 min at 4 °C) to pellet dead cell debris. At this stage, the supernatant will typically contain EVs of various sizes, which often enclose 20S proteasomes and activator subunits. To collect large- and medium-sized EVs, the resulting supernatant should then be centrifuged at 10,000–20,000× *g* (20–45 min at 4 °C) to pellet them [6,62,76]. 

To separate free and small-EV-encapsulated proteasomes, the sample should undergo a final step of ultracentrifugation at 100,000× *g* or higher [106]. The time and relative centrifugal force applied during this ultracentrifugation step are critical. For example, ultracentrifugation for 5 h at 200,000× *g* will sediment free 20S proteasomes [107], but proteasomes may also sediment at lower levels if a longer centrifugation time, e.g., overnight is employed. Therefore, extra care should be taken to ensure that free proteasomes are not collected together with the vesicles. Therefore, we recommend the use of a 100,000× *g* centrifugation step for no longer than 4 h [108]. Similarly, when adapting protocols using different types of centrifuges and rotors (e.g., if moving from a fixed angle rotor to a swing out rotor), it is necessary to perform proper rotor conversion calculations and consider the reported k-factor for each rotor in order to pellet the desired particles equally in rotors of different sizes and types [108]. 

In studies focusing on EVs, a series of other approaches is often employed in addition to differential centrifugation, such as separation on a density gradient, ultrafiltration, immunoaffinity-based purification, size-exclusion chromatography and polymer precipitation, some of which can further separate free proteasomes and those encapsulated in different sized EVs from each other [109,110]. Each of those methods has its advantages and disadvantages and should be selected with care, according to the specific experimental system and research objectives in question. 

After successful preparation of the body fluid and EV being examined, the sample should first be monitored for proteasomal catalytic activities triggered by ChL, CasL and TrpL. Various site-specific fluorescence-based peptides have been developed over the years, which are detectable upon specific cleavage by each of the three catalytic subunits of the proteasome [111]. In addition, various protein substrates such as α-synuclein, the C-terminal domain of p53 and oxidized calmodulin can be employed to monitor the degradation of intact proteins by 20S proteasomes [112]. Similarly, several fluorescence-based protein substrates such as ODC-GFPSpark [113] and GFP-α-synuclein [114] are also available. GFP itself is resistant to degradation by the 20S proteasome; therefore, degradation of these substrates is performed by monitoring changes in size of the GFP fusion protein, either by means of fluorescent native gels or Western blot analyses. GFP, on the other hand, is readily degraded by the 26S proteasome upon ubiquitination. Therefore, if questions arise as to whether the sample contains active 20S or 26S proteasomes, degradation of ubiquitinated GFP can be monitored by measuring fluorescence decay [115]. In addition, other 26S proteasome-specific substrates are also available, such as ubiquitin-like domains or enzymatically ubiquitinated proteins tagged with fluorescent dyes, whose degradation is monitored by measuring the fluorescence anisotropy [116].

A complementary approach for monitoring proteasome activity employs specific inhibitors, which bind in either a covalent or a non-covalent manner to the different catalytic sites of the proteasomes [117]. Some of the covalent inhibitors are also fluorescently labeled, and can serve as activity probes [118]. It should be noted, however, that when monitoring proteasome activity and inhibition, attention should be given to the option that other proteases in the sample may also cleave the activity peptide, giving rise to non-specific “false positive” signals. Similarly, it should be established that the solvent used to dissolve the substrate or inhibitor does not affect proteasomal activity, as we have noticed in plasma (our unpublished results).

Fluorescent activity- or inhibition-based probes are widely used in solutions. Such assays are typically performed in microplate wells and quantified using a fluorescence microplate reader [119]. However, due to the high stability and integrity of proteasomes, their specific activity or inhibition can also be monitored following separation in native gels [120]. In such experiments, the body fluid being examined is separated on a non-denaturing native polyacrylamide gel. Following separation, the gel is soaked in a solution containing a fluorescent probe, and after a short incubation, regions in the gel where active proteasomes are present, can be detected by a fluorescent camera. The spatial segregation of the different proteasome assemblies is particularly meaningful, since it enables not only monitoring of the activity or inhibition of the proteasome, but also determines the different assembly states of proteasomes in the sample being analyzed. Moreover, native gels can also be used in Western blot experiments probed with antibodies specific against different proteasome subunits to determine the presence of immunoproteasomes and various proteasome assemblies, as well as for relative quantification of different proteasome subspecies within samples. 

## 6. Discussion

The image of the extracellular proteasome that we present here by merging an array of pixelated data, gave rise to an unexpected discovery, namely that proteasomes are found in every body fluid we examined. In particular, subunits of the 20S proteasome and PA28α/β particles were identified, suggesting that free 20S proteasomes and 20S/PA28α/β complexes are the most abundant proteasome configurations in body fluids. Moreover, it appears that pathological conditions such as cancer and autoimmune disease lead to elevated proteasome levels in the extracellular milieu, suggesting that proteasomes in body fluids play a functionally relevant role(s) in physiological responses. It is not clear what triggers the presence of the proteasome in extracellular fluids and how the complex levels are coordinated, but it is possible that the inflammatory response, common to most pathological conditions, is the causative agent. Collectively, these findings challenge the current perception that the proteasome complex merely consists of intracellular machinery and call for a joint scientific effort aimed at understanding the mechanistic details in the physiology and pathology underlying extracellular proteasomes.

It is well-known that the 26S/20S proteasome ratio is highly dependent on the presence of ATP [121]. In the absence of ATP, the 26S proteasome rapidly dissociates into the 20S and 19S particles. This phenomenon may suggest that ATP-independent degradation by 20S and 20S/PA28α/β proteasomes, and not the 26S particle, constitute the dominant mode of proteolysis in the extracellular space, given that the extracellular concentration of ATP is ~1000-fold lower compared with that seen within intracellular fluids (between 3 to 10 mM ATP within cells, and only about 10 nM in the extracellular space) [122,123]. Other factors that may explain the dominance of 20S and 20S/PA28α/β proteasomes in the extracellular milieu are the simplicity and self-reliance of the system. Unlike the 26S proteasome degradation route, 20S and 20S/PA28α/β proteasome-mediated degradation does not involve a sophisticated enzymatic cascade involving three different types of enzymes (E1, E2, and E3) that ubiquitinate substrates in an ATP-dependent manner. Nevertheless, it should be noted that there remains a level of ambiguity regarding the configuration of proteasomes in extracellular fluids. Although in plasma it indeed seems that the 20S proteasome is the dominant particle [10], it is not clear whether this is also the case in other extracellular compartments. Therefore, there is an urgent need to develop standardized assessment methods that distinguish between the two proteolytic systems.

PA28α/β is the dominant proteasome activator found in body fluids (Figure 2 and Figure 3). It is known that during infection, PA28α/β and the 20S immunoproteasome subunits are induced by IFN-γ to form 20S/PA28α/β immunoproteasomes, facilitating the generation of MHC class I ligands for subsequent antigen presentation [26,124,125,126]. The peptide fragments generated by 20S/PA28α/β immunoproteasome particles differ from those formed by the 26S proteasome complex, suggesting that PA28α/β is involved in adjusting the antigen presentation or in controlling excessive cytotoxic responses against self-originating antigens by preventing autoimmune reactions [127]. Thus, it is reasonable to speculate that extracellular proteasomes in (some) extracellular fluids contribute to the overall immune response against extracellular pathogens. Dendritic cells, for example, are antigen-presenting cells localized in organs exposed to the outer environment; namely, the skin, lymphoid organs, blood and mucous membranes in the respiratory and digestive tracts. Furthermore, dendritic cells specialize in presenting exogenous antigens that are not internalized, on MHC I [128,129]. One route for presenting peptides from extracellular origins is through cross-presentation, by which extracellular antigens are internalized by the dendritic cells, where they are transported from the endosomal compartment into the cytosol and degraded by the proteasome. The resulting peptides are then transported back into the ER or to antigen-containing endosomes, where they are loaded onto MHC class I molecules for antigen presentation [130]. The 20S/PA28α/β immunoproteasomes may take part in an extracellular defense mechanism and generate antigenic peptides outside of the cells. These peptides may then be internalized through cross-presentation and directly presented by MHC I molecules, without the need for degradation by cytosolic proteasomes. 

In sweat, for example, the proteasome was identified together with a range of other proteases and protease inhibitors, as well as several antimicrobial peptides, highlighting the intense proteolytic activity of human skin and the functional role of sweat in host defense and innate immunity [131]. This defensive role in protecting the body from external pathogens may possibly hold true in other body fluids such as tears, urine, saliva and cervicovaginal fluids, all of which are in close contact with the external environment. Overall, it is clear that our current understanding only touches the “tip of the iceberg”, and that to understand the functionality of the extracellular proteasome in-depth research is required.

In general, proteolytic degradation is part of the proteostasis network that regulates the proteomes [132]. Proteostasis involves an interplay between various pathways that influence the fate of a protein from synthesis, ranging from folding and function to degradation. The finding of proteasomes in the extracellular milieu suggests that a protein quality control system also operates in the extracellular spaces of multicellular organisms. This view is supported by the discovery of extracellular chaperones that share functional characteristics with intracellular small heat shock proteins [133], and by the fact that about 13% of the human proteome is predicted to be secreted [134]. Moreover, a range of pathologies associated with extracellular protein deposition and aggregation exists, such as corneal dystrophies, atherosclerosis and glomerulonephritis [133], suggesting that the extracellular environment also depends on balanced proteostasis. The similarities and differences between the proteostasis network and its associated regulatory circuits in the extracellular space, compared to those of the intracellular system, is a topic that deserves careful attention. This is essential not only for a basic understanding of the extracellular proteasome, but also to enable identification of potential targets for future diagnostics and therapies.

In challenging the current dogma that proteasome-mediated degradation is limited to the intracellular environment, many questions are raised that need to be explored. For example, we do not know the origin(s) of extracellular proteasomes. Multiple scenarios may explain their extracellular presence, such as active transport from the intracellular into the extracellular milieu, release through encapsulation within extracellular vesicles and passive release from ruptured cells. Considering the array of body fluids that display proteasomes, we anticipate that a single mechanism cannot explain the presence of them all. Rather, a combination of processes takes place, depending on the body fluid compartment in which the proteasomes are found, as well as the pathophysiological state. In this regard, we also lack understanding of the mechanisms that regulate proteasome levels, how extracellular proteasome concentrations are related to disease etiology and the nature of the clearance mechanism(s) and half-life of these complexes.

Likewise, the specific physiological function(s) achieved by extracellular proteasomes is not known, as well as their synergistic involvement in the proteostasis network. They may, for instance, share functional characteristics with intracellular proteasomes and clear damaged and unfolded extracellular proteins, thereby protecting cells and tissues from the toxic or physically disruptive effects of protein aggregates. They may also accelerate the immune response by generating antigenic peptides, or may actually act differently, in an as-yet unknown manner, than their intracellular counterparts. We also do not yet know the substrate pool of distinct extracellular proteasomes, their specific source origins and how and if these change with aging. The exact composition of extracellular proteasomes is also unclear. Diverse populations of distinct extracellular proteasome entities may exist [135], each harboring subunits with different post-translational modifications, splice variants, alternative initiation sites and cofactors. It is, therefore, essential to propel the field forward through comprehensive investigations of the molecular details that govern and coordinate extracellular proteasome functions. 

The increasing pace of activity expected in this field urgently calls for the development of dedicated biochemical methods and analytical tools. For instance, one of the challenges in studying this system is that cell-line-based research, which previously laid the foundations for understanding intracellular degradation, cannot sufficiently reflect the physiologically relevant context of extracellular proteasomes. This is particularly significant in studies of extracellular proteasomes seen in different pathophysiological states, and coupled with the system’s adaptive immune response. Therefore, in order to gain insights into the workings of extracellular proteasomes, there exists a need for appropriate animal models. To that end, we expect that transgenic mice strategies will enable both the monitoring and isolation of extracellular proteasome particles. Moreover, pharmacological modulation of current proteasome inhibitors to cell-impermeable compounds will likely facilitate the selective inhibition of extracellular species, paving the way towards unraveling the substrate repertoire and its functional significance. Considering the challenges inherent in distinguishing between the 26S and 20S proteasomes, as well as the ambiguity regarding the presence of various proteasome activators in the extracellular milieu, the availability of new tools that accurately report on the presence, integrity and activity of specific proteasome configurations (e.g., free 20S vs. 20S bound to an activator, such as the 19S or PA28α/β) will greatly facilitate our understanding of the complex biological roles of proteasomes in extracellular fluids. Finally, it is expected that as the field matures, standardized methodologies used to quantify the abundance, heterogeneity and activity of extracellular proteasomes will be established. 

The importance of the in-depth exploration of extracellular proteasome degradation pathways also lies in the potential for diagnosing and monitoring disease. Increased levels of proteasomes have been detected in EVs, as well as in circulating in plasma. Moreover, disease prognosis and survival rates are correlated with extracellular proteasome levels in plasma. Accordingly, effective therapies have been shown to lead to reductions in extracellular proteasome particles. Thus, it is possible that evaluation of distinct body fluids, especially blood plasma, will enable the use of levels or enzymatic activities of proteasomes to predict clinical states. Moreover, the specific role(s) of extracellular proteasomes should be fully determined before therapeutic approaches can be entertained. For example, it remains to be determined whether influencing extracellular proteasome levels impacts a pathophysiological state, or whether its levels constitute a side effect of the condition itself. Although the road is long, we anticipate that a deeper biological understanding of the extracellular proteasome system, alongside development and validation of precise, reliable and robust tests to clarify its functioning, will point towards new avenues for disease diagnosis and therapies.

## Figures and Tables

**Figure 1 biomolecules-12-00619-f001:**
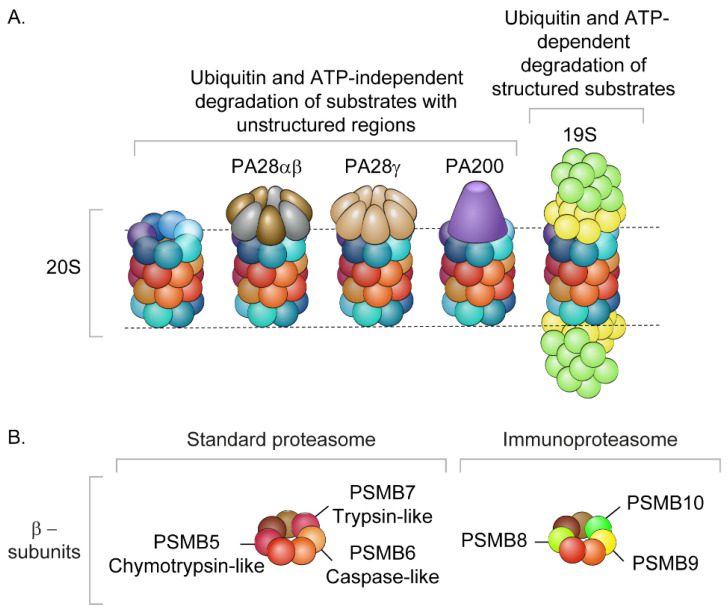
Two alternative proteasomal degradation mechanisms, which are not mutually exclusive, exist in cells. The first is ubiquitin- and ATP-independent, while the second is dependent of both. (**A**). Ubiquitin- and ATP-independent degradation can be undertaken by the free 20S proteasome complex. This complex is composed of 28 subunits arranged in a cylindrical structure comprising four heptameric rings: two outer α-subunit rings (PSMA1–PSMA7, colored in cool colors) that embrace two central β-subunit rings (PSMB1–PSMB7, colored in warm colors). Alternatively, the 20S complex can be capped at one or both ends by the proteasome activators; PA28α/β, PA28γ, and PA200. PA28α/β (gold and silver) and PA28γ (beige) form hetero-and homo-heptameric rings, respectively, while PA200 (colored in purple) is a monomeric protein. On the other hand, ubiquitin and ATP-dependent degradation is performed by the 26S and 30S proteasomes that are formed by binding of one or two 19S regulatory complexes to the 20S proteasome, respectively. Two substructures form the 19S regulatory particle: the lid (colored in green) and the base (yellow). Hybrid proteasomes can also be generated when the 20S proteasome is capped with two different activators, mainly the 19S particle, and either PA28α/β and or PA28γ. (**B**). Three of the 20S proteasome β-subunits, PSMB5, PSMB6 and PSMB7, are catalytically active. The immuno-proteasome contains three alternative catalytic subunits: PSMB8, PSMB9 and PSMB10.

**Figure 2 biomolecules-12-00619-f002:**
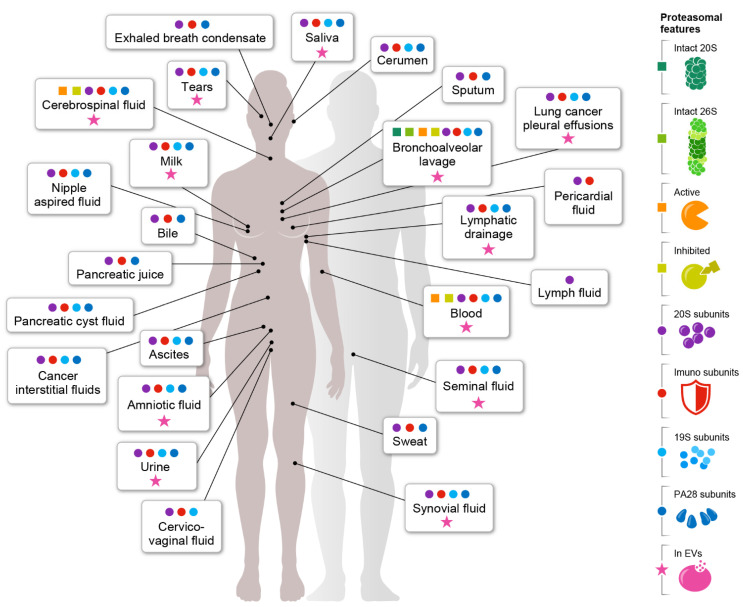
Proteasome subunits are found in at least 25 body fluids. Analysis of 143 proteomic datasets revealed that proteasome subunits are found in 25 different body fluids. The schematic map highlights the different body fluids, which indicate the presence of 20S proteasome subunits (purple circles), immunoproteasome subunits (red circles) and subunits of the 19S (light blue circles) and PSA28αβ activators (dark blue circles). In addition, the scheme indicates whether intact 20S (dark green squares) or intact 26S proteasomes (light green squares) were found, whether the activity of the proteasomes was determined (orange squares) and if it was inhibited by proteasome inhibitors (lime green squares). Pink asterisks indicate body fluids in which proteasome subunits were found within extracellular vesicles.

**Figure 3 biomolecules-12-00619-f003:**
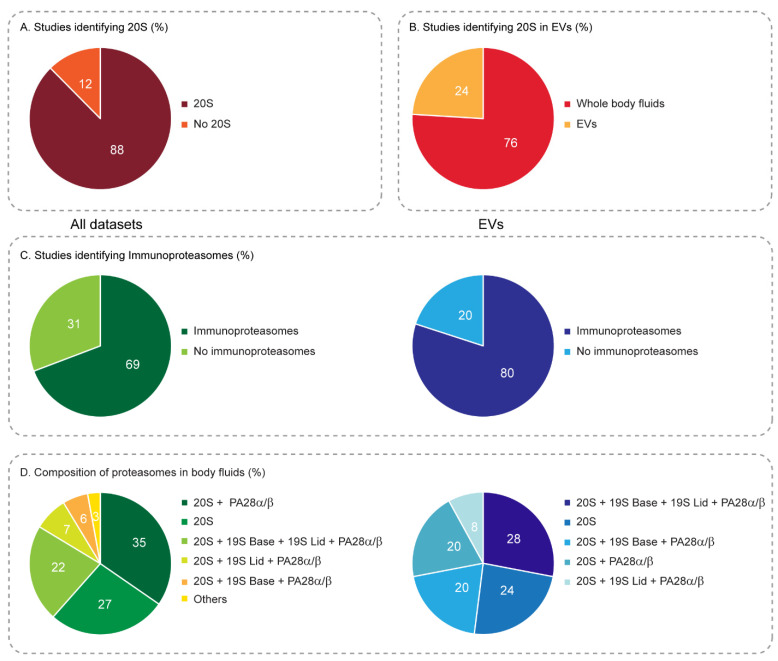
Overall characteristics of proteasome subunits in extracellular fluids. (**A**) Overall, 143 proteomic datasets of body fluids were examined. Proteasome subunits were identified in 88% of these studies. (**B**) Here, 76% of the studies were performed on whole body fluids and 24% were performed on body fluid EVs. (**C**,**D**) Percentage of studies identifying at least six 20S proteasome subunits, either in the overall body fluid (left panels) or enclosed within EVs (right panels). (**C**) Immunoproteasome subunits were identified in 69% of the datasets and in 80% of the EV-focused studies, suggesting that immunoproteasomes play a significant role in EVs. (**D**) Composition of proteasome subunits in the different body fluids (left panel) and in EVs (right panel). Other species denote datasets identifying the 20S + 19S base, 20S + 19S lid, 20S + 19S base + 19S lid, each found in 1% of the datasets.

**Figure 4 biomolecules-12-00619-f004:**
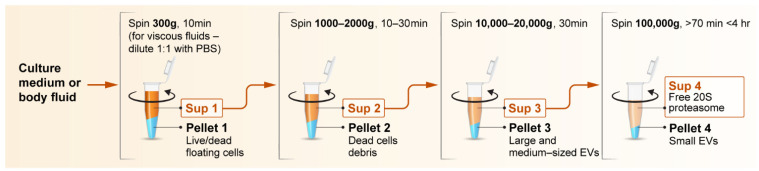
Recommendation of distinct centrifugation steps for handling body fluids and cell culture media directed towards proteasome investigation. Initially, the liquid should be centrifuged at 300–500× *g* to remove live and dead floating cells without rupturing them, to minimize spillage of their contents into the extracellular fluid. The supernatant (Sup) should then be centrifuged at 1000–2000× *g*, to pellet insoluble dead cell debris. At this stage, the cleared fluid will contain a variety of different extracellular vesicles, as well as freely floating proteasomes. Centrifugation at 10,000–20,000× *g* will pellet large- and medium-sized EVs, and another round of ultracentrifugation at 100,000× *g* will pellet small EVs. The final supernatant will contain free proteasome particles.

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
