# Peer review of "Biology of the Extracellular Proteasome"

_biomolecules, 2022, doi:10.3390/biom12050619_

Round 1
Reviewer 1 Report
The review “ Biology of the extracellular proteasome” by Gili Ben-Nissan, Naama Katzir, Maria-Gabriella Füzesi-Levi and Michal Sharon, is a very well written, exhaustive review on this frontier topic and I thus encourage its publication.
Author Response
We were very pleased that the reviewer recognized the significance and novelty of the review.
Reviewer 2 Report
This review article is significant and important to summarize and understand the extracellular proteasome.
I think this article is valuable to publish in Biomolecules after responding to several comments as follows.
1: The illustration of proteasomes should be more in detail.
The description of subunit composition and function of sub-complexes should be presented in figures. It would help wide readers.
2: The description of concrete substrates of the extracellular proteasome in each fluid is meaningful. Especially disease or physiologically significant substrate or contribution of extracellular proteasome should be explained if there are reports.
Author Response
- We thank the reviewer for this comment and have added more details to Figure 1 as suggested.
-
We agree with the reviewer, however, currently, there is no evidence for concrete substrates of the extracellular 20S proteasome. We hope this review will encourage research groups to join this field and bridge this gap in knowledge. In the manuscript, we emphasize this aspect:
Page 9, second - “Nevertheless, much knowledge is lacking concerning the nature of plasma proteasomes, as their origins, mode of regulation, composition and substrate specificities in physiology and pathology remain unclear”.
Page 17, third paragraph – “We also do not yet know the substrate pool of distinct extracellular proteasomes, their specific source origins, and how and if these change with aging”.
Reviewer 3 Report
The ubiquitin-proteasome system represents the main pathways for targeted protein degradation in the cell. The proteasome is a multi protein complex that acts as a molecular shredder by degrading ubiquitin tagged proteins into small peptides. Historically the proteasome was assumed to by cytosolic, however it is known know that a large component of the proteasome fraction exists in the nucleus. The existence of extracellular proteasomes has been known for several years however the significance of this is relatively poorly understood. In this review article the authors shed much needed light on the existence and potential function of extracellular proteasomes, including the presence of complex hybrid species containing alternative activators and regulators. Overall this is quite simply a perfect review article. The authors should be commended on not only writing a comprehensive and well researched review, but also one that is engaging and fascinating in its content. I wholeheartedly agree with the authors that it will '' stimulate thoughtful discussions on this exciting topic''. I recommend immediate publication.
one minor quibble
Figure 1. Two alternative proteasomal degradation mechanisms, which are not mutually exclusive, exist in cells. The first is ubiquitin and ATP-dependent, while the second is independent of both.
I would change the order of the sentence so as to reflect the order of proteasome figures.
Author Response
We thank the reviewer for this comment, which has been corrected accordingly.